# Law and (rec)order: Updating memory for criminal events with body-worn cameras

**Delene Adams, Helen M. Paterson**⬦\*, **Hamish G. MacDougall**

School of Psychology, University of Sydney, Sydney, NSW, Australia

\* helen.paterson@sydney.edu.au

## Abstract

Body-worn video is increasingly relied upon in the criminal justice system, however it is unclear how viewing chest-mounted video may affect a police officer's statement about an event. In the present study, we asked whether reviewing footage from an experienced event could shape an individual's statement, and if so, whether reporting before reviewing may preserve an officer's original experience. Student participants ($n = 97$) were equipped with chest-mounted cameras as they viewed a simulated theft in virtual reality. One week later, half of the participants recalled the event in an initial statement while the other half did not. Participants then viewed either their body-worn video or a control video. Finally, participants provided their statement (no initial statement condition) or were given the opportunity to amend their original account (initial statement condition). Results revealed that viewing body-worn video enhanced the completeness and accuracy of individuals' free recall statements. However, whilst reviewing footage enabled individuals to exclude errors they had written in their initial statements, they also excluded true details that were uncorroborated by the camera footage (i.e., details which individuals experienced, but that their camera did not record). Such camera conformity is discussed in light of the debate on when an officer should access their body-worn video during an investigation and the influence of post-event information on memory.

## Introduction

It is commonly believed that we encode and retrieve memories similar to the way a camera records and replays video footage [1, 2]. Consequently, eyewitness testimony can be one of the most persuasive forms of evidence for juries [3]. In fact, the only evidence more powerful than "I saw it with my own eyes" is "I have it recorded on camera" [4, 5]. For this reason, body-worn cameras were introduced to police officers' uniforms to document public encounters with levels of objectivity, transparency and accuracy unattainable with memory alone [6]. This has prompted the question: why rely on an officer's memory when events can be seen on video? Since very little is known about the effects of body-worn video on memory, researchers have highlighted the need for further research in this area [7]. The aim of the present study is to investigate whether there are differences between how an individual remembers an event and the details caught on camera.

**Competing interests:** The authors have declared that no competing interests exist.

Body-worn cameras have been rapidly implemented in police forces worldwide as a proposed solution to a police-legitimacy crisis, most notably in the U.S. [8]. This follows an increase in citizens documenting cases of police misconduct using mobile phones, which has sparked media interest, public scrutiny and protests. Body-worn cameras now offer a more standardised procedure of recording police-citizen interactions to protect the public against police misconduct and to protect police from false allegations and unfounded complaints. Video holds value for prosecutorial outcomes, whereby failure to secure convictions often comes down to the strength of evidence [9, 10]. Similarly, defence attorneys can view legal violations onscreen or ascertain whether an officer's testimony is corroborated by their footage. Ultimately, video 'tells us exactly what happened' [11]: an ideology which extends throughout the justice system, echoed in one case heard by the U.S. Supreme Court. In this case, judges allowed camera evidence to "speak for itself" by uploading police dash-cam footage online to uphold their ruling on an officer's use of force (Scott v. Harris, 2007).

The potential for body-worn cameras to improve evidence quality in criminal investigations is cited as one of the main reasons for the rapid diffusion of these devices worldwide [12, 13]. However, commentators have noted that the technology has advanced faster than empirical evidence can inform policies governing the use of the devices [7, 14, 15]. Consequently, there are currently no evidence-based guidelines to inform police forces on best practice of body-worn video use within report-writing [7, 16]. In light of several forces mandating their officers record all citizen encounters, body-worn video can quickly prove costly in terms of data storage and hours officers must spend reviewing incidents [15]. Therefore, whilst body-worn cameras may be an effective tool to enhance policing, how these tools are implemented may influence their benefits and thus warrants investigation. An exploration of the different reviewing and reporting practices is one area that would facilitate incident-reporting.

One controversial debate in policy revolves around an officer's access to the footage they collect. Should police officers always review this footage before writing an official statement? Or, would an initial statement written prior to viewing footage better reflect their experience? Below we outline arguments for each case, followed by a description of the current study.

## Support for officers viewing video footage prior to statement-writing

It makes intuitive sense that police officers should access all the available evidence before forwarding the case to the prosecution. Indeed, in data collected on fifty of the largest police departments in the U.S., the majority grant unrestricted access to footage, even where an officer's own actions are called into question [17]. This recognises that, unlike a recording device, individuals cannot encode every detail within their visual field, nor can they later retrieve all the information they encode. A police officer must react to situations that are unpredictable, dangerous and complex. They are tasked with split-second decision-making, de-escalating threats, apprehending suspects, recording information, protecting the public, as well as considering the safety of themselves and their colleagues. The nature of this role can result in impaired perception or memory for details of their encounters due to heightened stress [18, 19], physiological arousal [20, 21], cognitive demands [20], the presence of a weapon [22], or discharging their own weapon [23]. Therefore, an officer's expectations or intentions may cause them to misinterpret or even fail to process a detail that appears visually salient on their video footage [24, 25]. Memories of these details may then be subject to further impairment following a period of delay [26–31]. This may result in reports that omit forensically important details or include details that did not occur, which can bring about serious repercussions for that officer. Review, therefore, allows officers to update their memories, so that they can provide detailed incident reports that are free from error.

Whilst psychological literature has yet to address the effects of reviewing body-worn video on memory, the impact of photos capturing an individual's day has been investigated with devices such as *Sensecam* [32]. Sensecam is a device worn around the neck that takes a continuous stream of photographs or is triggered by sensors detecting changes in light, temperature and movement. Review of these photographs can increase the number of details wearers recall about events in that day, an effect that also persisted after a delay and for events not shown in the images [33, 34]. Such studies are, however, less suitable for verifying the accuracy of memory beyond details within the photographs, and authors have suggested the distinctiveness of wearing a camera on trial days can contribute to enhanced recall for events during those days [34].

Photographs have proven powerful retrieval cues for episodic memory within individuals with memory impairments and cognitively healthy individuals (see [35] for a review). Viewing a photograph can strengthen a memory trace for an event via retrieval practice or rehearsal prior to an individual being tested [36–38]. Moreover, a photograph's rich perceptual sensitivity and personal salience may facilitate the reconstruction or reactivation of a more coherent representation [39, 40]. In line with this, individuals have frequently reported that reviewing photographs have evoked feelings of 'reliving' that event [41–45], rather than merely 'knowing' something has occurred because the photograph says so [41, 46]. Similarly, body-worn video enables an officer to mentally return to the scene of an incident, reinstating the context in which they originally encoded various memories for that event (encoding specificity principle; see [47]). As body-worn video is self-generated [48] and shares a significant overlap with the original encoded memories of the officer recording the event [49], it is likely to be a highly effective retrieval cue for episodic memory.

Body-worn video may not only strengthen the activation of details within memory, but also details these are bound to or associated with beyond what the camera has recorded. For instance, spreading activation theory describes memory as being organised in a distributed and associative network [50, 51]. Therefore, a visual detail viewed onscreen will be activated in memory where it has been encoded within its spatial context (location, what was nearby), temporal context (timing within a sequence of events, what occurred before and after), social context (who was present), cognitive context (thoughts, motivation for actions) and emotional context [39]. The co-activation of different components within the event increases the likelihood of further components being activated and bound together to form a single coherent episodic memory [52, 53]. Without review, these additional details may have been inaccessible or below a threshold required to be retrieved from memory [33]. This suggests that reviewing body-worn video may elicit a more complete episodic memory trace for an experienced event, facilitating retrieval of details on and off-screen.

Research investigating the impact of body-worn cameras on memory within a police setting is limited to one study that explored whether video facilitated incident-reporting following a use-of-force training scenario [54]. Eleven officers provided initial statements before reviewing their eyewear footage and subsequently amending their statements. Results indicated more accurate incident-reporting as a result of officers eliminating errors, albeit not entirely, written in their original reports. Similar results have been found in trials of body-worn cameras with paramedics in the documentation of an emergency callout [55]. However, absent from both of these studies are samples large enough to determine whether there are statistically significant differences in memory outcomes under different statement-writing and review conditions. Furthermore, an understanding of the influence of video on memory after a period longer than the fifteen-minute delay implemented in Dawes et al. [54] remains to be explored. Nonetheless, these initial findings emphasise the value of body-worn video in providing additional information to investigations. Importantly, what an officer can recall from an incident may largely diverge from their camera's recording.

## Support for officers providing an initial statement prior to viewing video footage

On the other hand, researchers propose treating an officer's testimony as an independent source of evidence; one that is not contaminated by other sources, including camera footage [16, 56, 57]. A minority of police forces implement this policy and prohibit review until officers have written a report or have been questioned about an event, especially in officer-involved shootings or alleged misconduct [17]. The main concern regarding unrestricted access to camera footage is that officers may extract details for their statement solely from the camera footage, rather than their own recollection of events. Whilst this may enhance accuracy, this is less helpful in cases where we want to understand how an officer remembers the event and their perceptions as the action unfolded: details that may not be visible on their chest-mounted camera. Put another way, an officer's statement ought to supplement video with the context and clarity of the story they witnessed, rather than the one recorded by the camera: a story that can "speak for itself" (Scott vs. Harris, 2007). At the extreme end of public mistrust, there are also concerns that officers can use the content shown or not shown on their recording to fabricate details in their statement.

There are a number of factors that complicate treating footage as identical to an officer's visual or cognitive experience during an incident [58]. Firstly, body-worn cameras are only of evidentiary value when switched on and for the entirety of the event. Even when activated, the reality is that body-worn cameras produce fragmented or blurred clips bound by what is in front of an officer. Then, chest-mounted cameras will not reveal the various directions in which an officer points their head, nor will they show the officer's own actions. This is noteworthy given that mounting cameras on an officer's chest is the most common location opted for by police departments [15]. An officer who turns their head may not see, let alone report images captured on their body-worn camera. Alternatively, they may report details outside their camera's field-of-view. Therefore, footage may overwrite an officer's memory or at the minimum, shape the details they choose to report.

There are also theoretical arguments supporting claims for an initial statement to be composed prior to reviewing footage. Research proposes that post-event information can alter an individual's original experience encoded in memory, especially following a delay [28]. Theories of reconsolidation note that whilst reactivation of an event in memory through re-exposure can enhance recall, it can also render that memory susceptible to modification [59] and distortion (misinformation effect; [60–62]). Similar to other witnesses or the media, body-worn video provides an alternative perspective to the event they attended. However, the specific content shown on video may determine precisely what is recalled or forgotten [63, 64]. Therefore, the power of video evidence to rewrite memory leads us to question whether body-worn video can cue memory for details outside the camera's field-of-view, or instead, make these details less likely to be reported.

One study that points towards the fragility of off-camera memories found that reviewing photographs on a single occasion produces subtle memory impairment for non-reviewed items relative to reviewed items for a previously watched video [52]. This finding was not robust across all of Koutstaal et al.'s [52] experiments, which is unsurprising given that photographs were reviewed for 20 seconds and recall was tested ten minutes after viewing the original stimuli. It is possible that a longer delay (such as those typically expected in criminal investigations) may exacerbate this effect, increasing an officer's reliance on the camera to remember events for them (transactive memory: [65]). For instance, a recent study required individuals to edit a photograph by cropping its parameters then review this edited version several days later. The findings revealed participants simultaneously experienced enhanced

recall for intact details yet impaired recall for removed details during a final recall test one week after taking the original photograph [66]. Therefore, following delay, the content of body-worn video may prompt both remembering and forgetting.

Turning to video more specifically, research highlights that the perspective of footage may impact how an individual interprets events [67, 68]. In Lassiter and Irvine's [67] study, participants rated a criminal interrogation as less coercive when watching a video showing only the suspect (i.e. from the interrogator's viewpoint) than when both interrogator and suspect were equally in shot or when reading a transcript of the interrogation. A similar "camera-perspective bias" was found with body-worn video, in which participants rated the intentionality behind a police officer's actions as less when viewing body-worn footage (omitting the police officer) than dash-camera video showing a third-person perspective of the same incident [69]. These studies suggest what is salient or caught on camera may be attributed a larger causal role in explaining events, relative to details outside the camera's viewpoint. Therefore, an officer's memory for an event they witnessed may be reduced to what can be verified by camera footage.

Taken together, our understanding of how body-worn video influences the reporting of events is limited to a small-scale trial of eyewear devices involving eleven officers [54]. This study provides crucial first steps in demonstrating that video can reduce erroneous recall. However, scientific evidence is needed to support and extend these conclusions with a larger sample and adequate control conditions. In other studies, individuals have been asked to make judgements based on pre-recorded body-worn video of events they have not experienced first-hand [56, 69]. Moving forwards, we address how body-worn video impacts memory for experienced events beyond the number of errors reported, following a delay, and seek to understand the relevance of these findings to chest-mounted cameras. For instance, it remains unknown whether reviewing chest-worn footage can prompt memory for details outside the camera's recording range or, instead, restrict memory to details visible onscreen.

## The current study

In the present study, we sought to understand whether, and if so how, camera footage updates memory for an experienced event. We immersed participants in a virtual environment in which they witnessed a crime. Participants were equipped with a chest-mounted device to record events, analogous to a body-worn camera used by a police officer. One week later, half of the participants were asked to record an initial account of the event while the other half were not. Next, half of the participants viewed their body-worn camera footage while the other half viewed a control video that was unrelated to the study. Finally, participants provided their statement (no initial statement condition) or were given the opportunity to amend their original account (initial statement condition). We predicted that participants who reviewed their body-worn video would produce statements which were more complete (total number of details) and accurate than participants who did not review their footage (Hypotheses 1 and 2 respectively). However, it was unclear whether this would hold true for all details, even those falling outside the camera's view. To address this, we tracked head and chest position to demonstrate that what individuals witness and what is caught on a chest-mounted camera are not one and the same. We predicted that participants who reviewed their footage would be less likely to include off-camera details, except for those who provided an initial statement before review (Hypothesis 3). That is, we believed the opportunity to prepare an initial statement may preserve any "off-camera" details participants could recall before they were shown what they had recorded on camera for the first time. These hypotheses aimed to shed light on a current debate surrounding when an officer should engage in recall of an event, before or after

reviewing footage. Proposals indicating that officers should engage in report-writing prior to footage review should be mindful that memory will inevitably appear inconsistent. Therefore, for those who were able to update their report, we predicted that those who reviewed their footage in between would produce more inconsistent statements than those who did not review their footage (Hypothesis 4).

## Method

### Participants

Ninety-seven participants (64 female, 32 male, 1 identifying as 'other') were recruited via an undergraduate online signup system and participated in exchange for course credit. Participants' age ranged from 17 to 42 years ($M$ = 19.90 years, $SD$ = 3.25). Those with a history of motion sickness or anxiety disorder were advised not to sign up for the study. The study received ethical approval from the University of Sydney Ethics Committee.

### Design

The study used a 2 (initial statement vs. no initial statement) x 2 (review vs. no review) between-subjects design. Recall data was collected in the second session, where participants were randomly allocated to one of four conditions (Fig 1). In this session, half of the participants were assigned to first provide an initial recall statement (initial statement condition) whilst half did not (no initial statement condition). Additionally, half of the participants were randomly assigned to review their chest-mounted footage (review condition), whilst half viewed a control video (no review condition). After viewing either video, those in the 'initial statement' condition had the opportunity to amend and update their statement, whilst participants in the 'no initial statement' condition provided their recall statement for the first time only after viewing either video.

The dependent variables were completeness and accuracy of final recall statements, plus statement inconsistency for those in the initial statement conditions ($n$ = 49). Further, the initial statement condition allowed separate analyses to be run with the within-subjects variable of statement timing (initial statement, final statement).

### Materials

**Stimulus video.** We recorded a short video (approximately 2 minutes) depicting a fictional theft in a bar-setting using a VUZE 3D 360˚ camera (Humaneyes Technologies Ltd. 9085000 Israel) positioned at the centre of the filming location. This allowed participants to watch the video from a first-person perspective in virtual reality. The video was presented in stereoscopic format, showing a different image to each eye which resembles binocular vision. This allowed us to create depth, with objects or actors appearing nearer or further away to the participant rather than equidistant. The video content involved a social gathering which culminates in two male perpetrators entering the scene, stealing a female victim's purse and personal items before fleeing. Other subtle criminal activity related to drug-use is visible, but no use of weapons or violence occurs. The full stimulus video can be viewed on the open science framework (https://osf.io/xbtvf/).

**Virtual reality.** A Vive VR headset (HTC, Xindian, New Taipei, Taiwan) with a field-of-view of 110˚ was used to display the 360˚ stimulus video in NeosVR software (Solirax Ltd. Prague, Czech Republic). A motion-tracked handheld controller functioned as a camera within the virtual environment, recording events with a 60˚ field-of-view in 720p resolution. This controller was attached to a vest and worn by participants so that it recorded from chest-

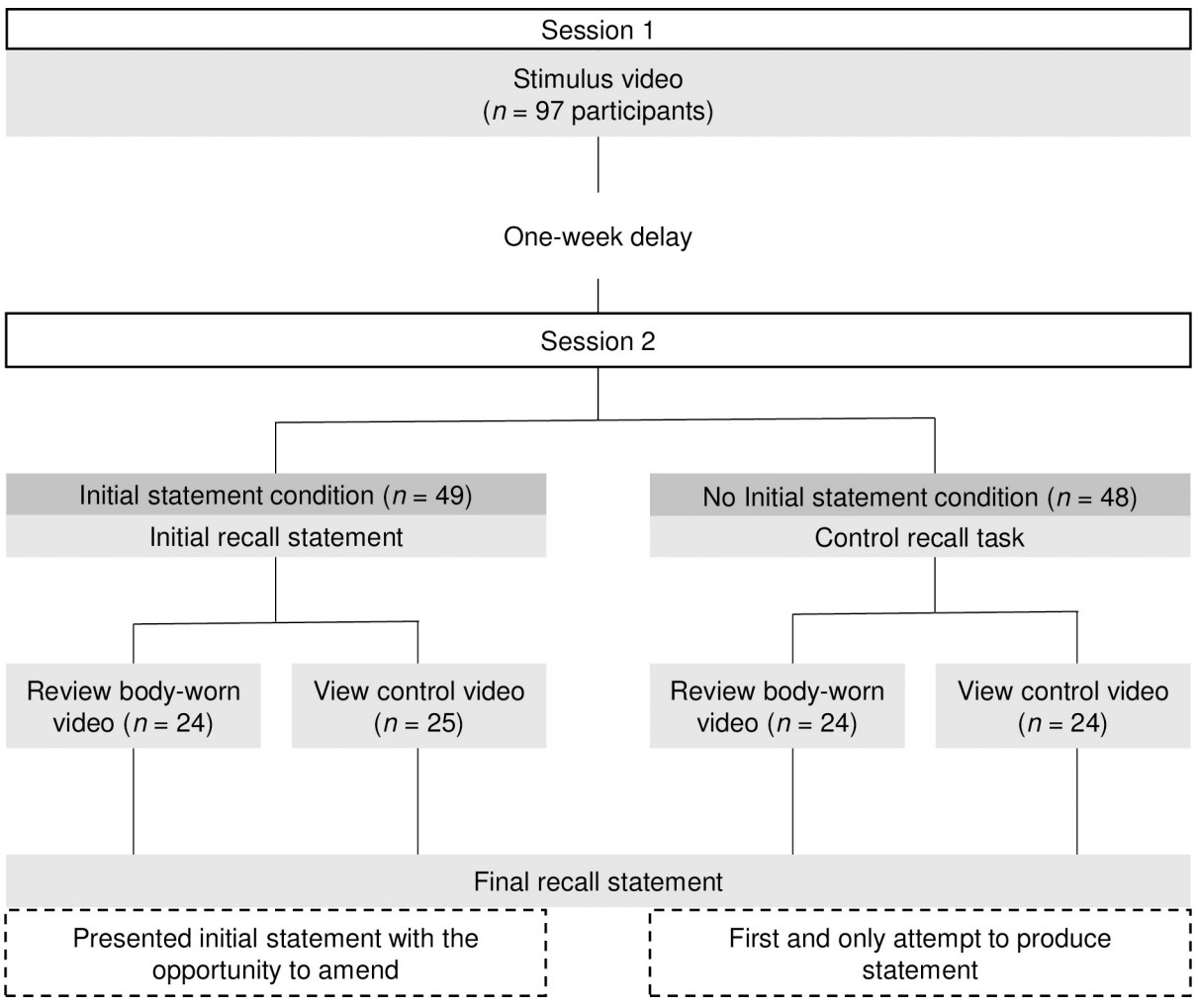

**Fig 1. Study design.**

height. Two infrared laser emitter units (HTC SteamVR Base Stations) plus integrated inertial measurement units were used to track the position and orientation of both the headset and controller at an accuracy of less than 0.1˚ rotation and rate of three measurements per second. This data was used to calculate participants' head-chest discrepancy score, or the difference between head and body orientation during the video.

**Body-worn video recordings.** The screen-recording application OBS (Open Broadcaster Software) was used to create a video-recording from the perspective of the motion-tracked Vive controller. This footage was shown to participants in the review condition on a computer monitor.

**Igroup Presence Questionnaire (IPQ).** The IPQ [70] measured participants' subjective experience of presence to assess the ecological validity of the stimulus video shown through virtual reality. This questionnaire contains 14 items, with three subscales measuring spatial presence (e.g. "I felt present in the virtual space"), involvement (e.g. "I was completely captivated by the virtual world") and experienced realism (e.g. "the virtual world seemed more realistic than the real world"), plus one item with high loadings on all three factors measuring general presence ("in the virtual world, I had a sense of being there"). Answers were provided on a five-point Likert scale from 1 ("strongly disagree") to 5 ("strongly agree").

**Free recall.** A free recall template titled 'statement of police' was completed by all participants on the computer. The statement instructed participants to report as much information as they could remember regarding their police investigation the previous week, providing accounts that were as complete and accurate as possible but to avoid guessing; an instruction given previously by Hope et al. [20] to a police-officer sample. Prompts were provided to elicit person, object, action, setting and dialogue details.

**Statement-writing scales.** Six statements assessed variables including participants' confidence, motivation, and the content of their statements, plus how helpful they found the video they watched to writing their statement (see https://osf.io/xbtvf/). One additional item was included for those in the review condition ("The video was a true reflection of what I saw last week"). Agreement was rated on a scale from 1 (strongly disagree) to 5 (strongly agree).

## Procedure

The experiment involved two sessions, approximately one week apart. During the first session, participants provided informed consent in written format. They then completed a familiarisation phase with the virtual reality headset, in which they viewed an animated mountain landscape. The experimenter informed them they would remain standing in one spot, but that they could turn to look all around them. Then, participants provided demographic information and received task instructions to imagine they were a police officer involved in an investigation of an environment believed to be at risk for illegal activity. They were told they would later be required to produce a statement of any events that took place. Additionally, they were informed they would be equipped with a camera to record events before being fitted with a vest to which the motion-tracked controller, functioning as their body-worn camera, was mounted. Then, the experimenter played the 3D 360˚ immersive stimulus video which formed the virtual reality police investigation. Following the video, participants completed measures regarding presence and comfort within the virtual environment.

All participants returned one week later for their second session and were reminded of their hypothetical role as a police officer. Participants in the 'initial statement' condition began by completing their statement recalling details from the stimulus video whilst the 'no initial statement' condition reported everything they could recall from the mountain landscape scene they saw during the familiarisation phase.

Participants were re-randomised into a review condition, which determined the video they watched next. Participants in the 'review' condition were informed they would watch the chest-mounted footage they collected during the first session. They were instructed to watch this once, with volume and without pausing, fast-forwarding or rewinding. Those in the 'no review' condition watched an extract taken from a nature documentary of similar length to the stimulus video.

Following either video, participants in the 'initial statement' condition were presented with their statement again and told they could make any changes they felt necessary, but that this was not compulsory. Participants in the 'no initial statement' condition gave their statement of events for the first time at this stage, with the same instructions and template as those in the 'initial statement' condition. All participants then completed measures regarding their statements and the videos they watched before being debriefed and thanked for their time.

## Coding

Statements were deidentified and coded blind to review condition. There were 443 possible details in the video that could be reported. These details were classified under categories used

by Wright and Holliday [71]: action (94), person (195), object (64) and setting (69), with the addition of dialogue (21) and off-camera details. 'Off-camera details' were unique to each participant, where a score of one was given for each detail not featured on their chest-mounted footage.

Recall statements were scored for completeness, accuracy and for those in the initial statement conditions ($n$ = 49), inconsistency. Statement completeness was measured by the total number of details (both correct and incorrect) provided. Accuracy was then calculated for each participant by dividing the number of correct details (present in the original stimulus video) by the total number of details reported. For participants in the initial statement condition, statement inconsistency was coded using the scoring index from Theunnissen, Meyer, Memon, and Weinsheimer [72]. One point was scored for each detail removed (omissions), added (commissions), and where a direct change was made (amendments) from initial to final statement (e.g. changing "the perpetrator wore *blue*" to "the perpetrator wore *black*"). Additionally, the accuracy of these inconsistencies was coded (i.e. changing a correct detail to an incorrect detail and vice-versa). A statement inconsistency score was calculated by summing the total number of omissions, commissions and amendments.

Ten statements were randomly selected and independently coded by a second scorer, who was blind to review condition, to assess inter-rater reliability. Pearson correlations were computed for correct and incorrect details, statement inconsistency, plus 'off-camera' details in all statements. These ranged from .693 to .984 (all $p$'s < .05), reflecting a good level of agreement between the two coders [73].

## Results

### Preliminary analyses

One-way ANOVAs were performed to ensure there were no pre-existing differences between the four conditions. These revealed no significant differences in participants' age, delay between first and second session (which ranged from 6 to 8 days; $M$ = 6.88 days), or subjective rating of presence within virtual reality, as measured by mean IPQ score ($p$'s >.840). Additionally, a chi-square test of independence highlighted no significant differences in the distribution of gender between conditions, $\chi^2$ (6, $n$ = 97) = 4.80, $p$ = .570.

### Recall performance in final statements

To understand how body-worn video shapes memory for an experienced event, and whether effects are influenced by when a statement is written, our main analyses involved 2 (initial statement vs. no initial statement) x 2 (review vs. no review) between-subjects ANOVAs on participants' final recall statements ($n$ = 97). No significant interactions or main effects of the initial statement condition were found ($p$s > .319). However, analyses revealed a significant effect of review condition on the completeness of statements, $F(1,93)$ = 22.72, $p$ < .001, partial $\eta^2$ = .196, where those who reviewed their body-worn video included more details (both correct and incorrect) in their final statements than those who did not review. There was also a significant difference in statement accuracy between the two groups, $F(1,93)$ = 36.66, $p$ < .001, partial $\eta^2$ = .283, where participants who reviewed their body-worn video produced higher accuracy scores than those who did not review. These accuracy rates appear to be driven by reviewers both providing more correct details, $F(1,93)$ = 32.598, $p$ < .001, partial $\eta^2$ = .260, and fewer incorrect details, $F(1,93)$ = 28.122, $p$ < .001, partial $\eta^2$ = .232, than non-reviewers (Table 1). These findings also apply across all categories of detail reported, with the exception of setting details (Table 2).

**Table 1. Mean number of accurate and inaccurate details recalled in final statements.**

| | | Correct | Incorrect | Completeness | Accuracy |
|---|---|---|---|---|---|
| | | *M (SD)* | *M (SD)* | *M (SD)* | *M (SD)* |
| Review | Initial statement | 42.29 (12.45) | 0.79 (1.35) | 43.08 (12.26) | 98.01 (3.51) |
| | No initial statement | 47.08 (16.69) | 1.04 (1.43) | 48.13 (16.48) | 97.39 (3.85) |
| No Review | Initial statement | 29.00 (10.54) | 3.48 (2.49) | 32.48 (11.46) | 89.25 (7.23) |
| | No initial statement | 29.13 (13.58) | 2.88 (2.74) | 32.00 (14.54) | 90.91 (8.57) |

*Note.* All main effects of the review condition are significant at p < .05

## Off-camera details

'Off-camera' details (i.e., details that participants provided in their final statements which were not visible on their body-worn video) were analysed as a proportion of total correct details, noting that total correct details differed between reviewers and non-reviewers. A 2 (initial statement vs. no initial statement) x 2 (review vs. no review) between-subjects ANOVA was performed on the proportion of 'off-camera' details in final statements. Contrary to our hypothesis, there was no significant interaction between review and initial statement conditions ($p$ = .646). However, results revealed a significant main effect of review, $F(1,93)$ = 5.59, $p$ = .020, partial $\eta^2$ = .057. Specifically, participants who did not review their body-worn video included a higher proportion of off-camera details in their final statements, $M$ = 0.08 (0.10) than participants who reviewed their body-worn video, $M$ = 0.04 (0.06). Whilst statements written after reviewing body-worn video included fewer off-camera details, it does not appear that writing an initial statement prior to review prevents this from happening.

We further performed a 2 (statement timing) x 2 (review) mixed-design ANOVA on the proportion of off-camera details within initial and final statements. Results revealed a statistically significant interaction between review condition and statement timing, $F(1,47)$ = 14.95, $p$ < .001, partial $\eta^2$ = .241 (Fig 2). Simple main effects analyses indicated this interaction was the result of a significant effect of statement timing on the proportion of 'off-camera' details provided by reviewers, $F(1,23)$ = 14.62, $p$ < .001, partial $\eta^2$ = .389, which was not found for non-reviewers, $F(1,24)$ = 0.06, $p$ = .816, partial $\eta^2$ = .002. Specifically, reviewers included a larger proportion of 'off-camera' details in their initial statement compared to their final statement (Fig 2).

## Statement inconsistency

Between-subjects analyses were performed on overall statement inconsistency scores (i.e. the number of details changed) for participants in the initial statement condition ($n$ = 49), depending on their assigned review condition. First, a chi-square test was conducted to see if there was a difference in participants' decision to amend their initial statement. Results revealed a

**Table 2. Mean number of correctly recalled details for each category in final statements.**

| | | Action* | Person* | Object* | Setting | Dialogue* | Total* |
|---|---|---|---|---|---|---|---|
| | | *M (SD)* | *M (SD)* | *M (SD)* | *M (SD)* | *M (SD)* | *M (SD)* |
| Review | Initial statement | 14.71 (3.92) | 14.63 (6.70) | 6.00 (3.15) | 5.67 (2.88) | 1.29 (1.04) | 42.29 (12.45) |
| | No initial statement | 17.00 (5.27) | 15.83 (9.65) | 6.50 (3.71) | 6.17 (3.80) | 1.58 (1.14) | 47.08 (16.69) |
| No Review | Initial statement | 11.72 (4.68) | 8.32 (4.21) | 3.48 (2.10) | 5.00 (3.67) | 0.48 (0.65) | 29.00 (10.54) |
| | No initial statement | 10.21 (4.92) | 8.63 (5.48) | 4.04 (2.37) | 5.42 (4.21) | 0.83 (1.34) | 29.13 (13.58) |

*Note.*

* *t* values significant between *review* conditions at $p$ < .001.

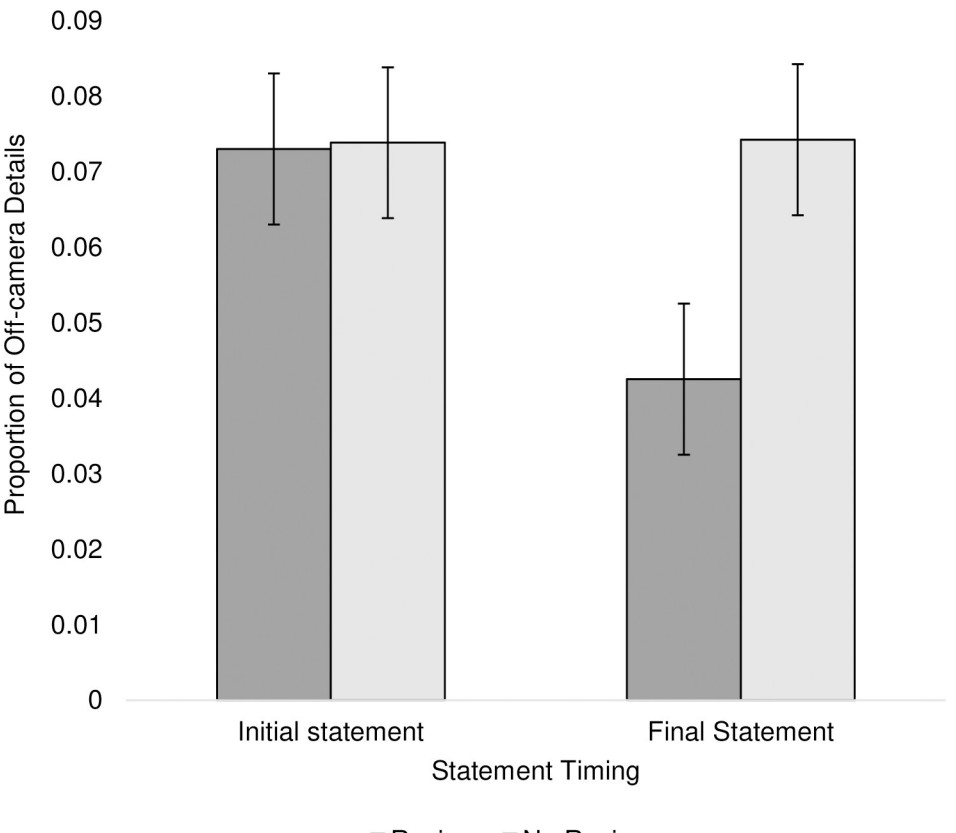

**Fig 2. Proportion of off-camera details participants included within their initial and final statements.**

significant difference between reviewers and non-reviewers, $\chi^2(49) = 12.06$, $p = .001$, where all participants reviewing their footage chose to make amendments compared to 60% of those who did not review their footage. A welch t-test was then run to compare statement inconsistency scores between reviewers and non-reviewers. This analysis was performed due to the violation of homogeneity of variances between groups, as assessed by Levene's test, $p < .001$. Results revealed a statistically significant difference between reviewers and non-reviewers' statement inconsistency score, $t(24.406) = 6.90$, $p < .001$, where participants reviewing their body-worn video altered more details in their initial statement than those who did not review. Exploring the different types of inconsistency revealed similar patterns, whereby reviewers included more omissions, $t(23.092) = 2.36$, $p = .027$, commissions, $t(25.392) = 6.38$, $p < .001$, and amendments, $t(28.309) = 3.68$, $p < .001$ in their final statements compared to non-reviewers. Further, reviewers' inconsistencies were more likely to be accurate; that is, omission of an incorrect detail, commission of a correct detail and amendment from incorrect to a correct detail (Table 3). There were no significant differences between reviewers and non-reviewers in relation to inaccurate inconsistencies, except for inaccurate omissions. That is, those who reviewed their body-worn video were more likely to omit correct details from their initial statements than those who did not review, $t(23.067) = 2.15$, $p = .042$.

### Head-chest discrepancy

We also calculated each participant's mean head-chest discrepancy score by subtracting their head orientation values from their chest orientation at each timeframe. Measurements were in

**Table 3. Mean statement inconsistency and accuracy scores (n = 49).**

| | Omissions | | Commissions | | Amendments | | Statement Inconsistency |
| --- | --- | --- | --- | --- | --- | --- | --- |
| | Accurate | Inaccurate | Accurate | Inaccurate | Accurate | Inaccurate | Total |
| | *M (SD)* | *M (SD)* | *M (SD)* | *M (SD)* | *M (SD)* | *M (SD)* | *M (SD)* |
| Review | 0.71 (1.23) | 2.29 (5.12) | 10.54 (7.71) | 0.50 (1.10) | 1.54 (1.77) | 0.04 (0.20) | 15.63 (10.04) |
| No Review | 0.04 (0.20) | 0.04 (0.20) | 0.64 (1.35) | 0.40 (0.71) | 0.12 (0.60) | 0.04 (0.20) | 1.28 (1.79) |

*Note*. 'Accurate' and 'Inaccurate' subheadings refer to the accuracy of the change from a participant's initial to final statement. Therefore, an accurate omission, for instance, would involve removing an incorrect detail.

degrees where the stimulus video could be mapped onto a 360-degree sphere in virtual reality. A root-mean square calculation was taken for all measurements. Participants' mean head-chest discrepancy score ranged from 15.21 to 66.65 degrees, $M = 38.37$ degrees (9.92). On average, participants' head was facing a direction of 38.37 degrees away from the orientation of their chest throughout the video.

Simple linear regression was conducted on a post-hoc basis to investigate the relationship between mean head-chest discrepancy score and memory inconsistency score for reviewers in the initial statement condition ($n = 24$). This was performed on the basis that the more participants' body-worn video diverged from their experience in virtual reality, the more statements may change following review. Results indicated that mean head-chest discrepancy score was not a significant predictor of memory inconsistency, $F(1,22) = 0.26$, $p = .617$, $R^2 = .012$. We then asked whether mean head-chest discrepancy score predicted the number of 'off-camera' details provided in participants' final statements, regardless of condition. This analysis revealed that head-chest discrepancy significantly predicted the proportion of 'off-camera' details reported in final statements, $F(1,95) = 27.44$, $p < .001$, $R^2 = .224$. Therefore, 22.4% of the variation in the proportion of 'off-camera' details can be explained by the model containing only head-chest discrepancy scores.

## Confidence

We performed a 2 (statement timing) x 2 (review) between-subjects ANOVA to assess participants' confidence in the details provided in their final statements. There was a statistically significant main effect of review condition on confidence, $F(1,93) = 9.80$, $p = .002$, partial $\eta^2 = 0.95$, where those who reviewed their body-worn video rated being more confident that the details in their final statement were accurate, $M = 4.15$ (0.77) compared to those who did not review, $M = 3.66$ (0.78).

We further performed a 2 (statement timing) x 2 (review) mixed-design ANOVA to understand whether there were differences between reviewers' and non-reviewers' confidence from initial to final statement. Our findings indicated that there was no statistically significant interaction effect ($p = .885$). Thus, it cannot be concluded from the data that participants experienced increased confidence after amending their statements in light of the body-worn video.

## Discussion

In the present study, we aimed to understand how reviewing video footage impacts memory for an experienced event and whether video content would shape the details recalled. In line with hypotheses, reviewing body-worn video enhanced both the completeness and accuracy of statements. Providing an initial statement prior to review did not influence either of these outcomes. We also found evidence that reviewing footage may shift a statement's focus towards

details recorded on camera, at the expense of details which individuals experienced but were not recorded. This is notable given our findings that participants' chest-mounted cameras did not capture everything they experienced. Contrary to our hypotheses, writing an initial statement did not preserve these off-camera details following review. Instead, these details appeared to be targets of removal when participants could amend their statement. Therefore, an initial statement written without access to footage may illustrate an officer's experience, rather than just the camera's recording, however, this did not prevent accounts of memory conforming to what was visible onscreen thereafter (a process we refer to as 'camera conformity').

The present study is the first to equip participants with chest-mounted cameras in a controlled laboratory setting and do so within the context of virtual reality (see [29, 74] for different approaches to eyewitness memory in virtual reality). This immersed participants within the environment in which the crime occurred, viewing events from a first-person perspective. Therefore, participants' experience depended on where they attended in the 360-degree environment. As in the real world, it was possible participants may not have noticed, nor their camera recorded, a criminal event at all, which differs to typical eyewitness literature in which participants are asked to passively view a crime scene on a two-dimensional screen. Our findings confirm that participants felt strong levels of both presence and involvement in the virtual environment. This mirrors findings in which individuals exposed to a 'bar-fight' in virtual reality reported higher feelings of presence and emotional responses (intention to aggress) than those reading a narrative of the same scene [75]. Both presence and emotion are important ingredients to forming an episodic memory [76]; the key variable of interest in the present study. Therefore, our research adds to the current knowledge of body-worn video by moving a step closer to real-world encounters from alternative laboratory-based methods, yet with the level of control and replicability unattainable in real-world settings.

As predicted, our findings confirmed the evidential value of body-worn cameras on statement-writing, with more details and fewer errors included in statements written with access to footage than without. This supports a large body of research which suggests that, following a delay, both the completeness and accuracy of eyewitness memory declines [77–79]. Yet, unlike most eyewitnesses, police often have access to video footage, which can counteract these effects. We provide further support for the findings that body-worn cameras can rectify inaccurate incident-reporting [54] in a controlled-laboratory setting in which participants were assigned to different reviewing and writing conditions. Moreover, we extend these findings to chest-worn video, the accuracy of details reported beyond the camera footage, and where a delay is incorporated between the witnessing and reporting of an event. These findings are also consistent with research showing that photographs from wearable cameras can facilitate episodic memory [32, 33]. Therefore, it may be asked why an officer should ever produce a statement without footage, where a visual record exists for an incident. However, it has been proposed an officer's experience, rather than their camera's perspective, is best captured in an initial statement prior to reviewing footage [16, 56, 57].

It is commonly believed that body-worn video *is* what an officer experienced [58], therefore we explored the overlap between an individual's experience and their chest-mounted recording by looking at the details attended to, yet not recorded by the camera. Our results support a suggestion made by Grady et al. [16]: video may shape the way we come to remember or report about an event. Specifically, participants who reviewed their footage were less likely to recall off-camera details than those who did not review their footage. However, contrary to our hypothesis, writing an initial statement did not preserve these details. We therefore turned to our findings on statement inconsistency to discover what happened when participants could amend their initial statements.

For participants in the initial statement condition, we expected those reviewing their body-worn video would produce a higher statement inconsistency score. Indeed, we found reviewers made changes to approximately fifteen details in their initial statement; a finding consistent with results obtained in a sample of police officers [54]. In contrast, those who did not access their footage altered one detail on average, with 40% opting not to make any content changes. The greatest benefit of review was the addition of details to statements. However, participants who reviewed their body-worn video and subsequently amended their initial statement did not show a significant increase in confidence from initial to final statement. In hindsight, it is possible that given the opportunity, participants may have rated their confidence in their initial statements lower after they reviewed their footage than they had done to begin with.

We also found that reviewing footage allowed participants to remove or amend any incorrect details. However, reviewers were also more likely to remove correct details than non-reviewers, with analyses revealing these to be off-camera details. That is, reviewing body-worn video causes individuals to filter out details not supported by the camera footage. This is important given that, on average, individuals faced 38.37 degrees away from where their chest-mounted camera was pointing throughout the stimulus video. Moreover, the greater the discrepancy between participants' head and chest orientation, the more off-camera details they reported. Therefore, whilst reviewing footage allows people to remove erroneous details from their accounts, it may also cause them to remove accurate details that they *did* experience. In a forensic context, this can result in duplication of what is already confirmed on-camera [57] and more concerningly, the potential loss of information that could be vital to a police investigation.

These findings lead us to question the impact of camera footage on episodic memory. Research suggests that an individual's memory for an event can be shaped by subsequent information they encounter (post-event information; [28]). Whilst body-worn cameras may provide the perspective of an 'independent witness' at an incident [80], unlike other witnesses, the details recorded and replayed by a camera will not be vulnerable to error. Therefore, it is likely that people will rely on recorded details given that the malleability of memory is driven by the credibility of the alternative source [81, 82]. Additionally, research indicates that individuals may treat cameras as an external hard drive for storing or offloading memories for events to begin with (transactive memory: [65, 83, 84]). Alternatively, it is plausible that rather than body-worn video changing episodic memory, participants merely altered their written statements to correspond with their footage. Therefore, the alternative perspective offered by an individual's footage may shape what they choose to recall. The result of this in the current study was that individuals omitted details they did experience to more closely align with the footage; a process of camera conformity that resembles memory conformity with co-witnesses [85, 86].

Our findings that individuals will omit details they personally experienced in light of body-worn video lend support to research conducted by Wright et al. [87], who highlighted that post-event information may not only result in the addition of non-experienced details, but the loss of experienced, albeit unrehearsed details. Further, Jones, Crozier, and Strange [56] found that participants were less likely to recall details included in an officer's written statement regarding use-of-force (i.e. the suspect carried a knife) when they had viewed disconfirming body-worn video. Similarly, our findings suggest that camera footage can be powerful at updating memory for an event, such that 'off-camera' details were edited out of participants' experience, with implications for policing that these details are not included within an officer's statement. Whether body-worn video shapes episodic memory or the wearer is motivated to achieve camera consistency, this adds a layer of difficulty in understanding an officer's in-the-moment perceptual experiences in order to rationalise their own actions (as per the direction within Graham vs. Connor, 1989).

In line with research on exposure to post-event information, we expected participants producing an initial statement in our study might prevent statements duplicating their camera footage. Specifically, this retrieval attempt may reactivate and strengthen the original memory trace for off-camera details [88], making these resistant to being overwritten by information the individual is exposed to thereon. For example, Hope et al. [85] found an initial statement protected police officers from incorporating erroneous details learned from conferring with other officers into their own accounts. According to discrepancy-detection theory [89], an initial statement can accentuate where an individual's memory diverges from new information sources. However, writing an initial statement without access to footage may reduce an individual's confidence in the accuracy of their memory, and therefore increase susceptibility to memory distortion [90, 91]. Therefore, proposals requiring an officer to produce an initial statement prior to reviewing footage acknowledges that their experiences and video will inevitably differ. However, our findings suggest using this initial statement to penalise an officer where discrepancies arise between memory and video is counterintuitive to this argument.

## Limitations and future research

We acknowledge that there are some limitations of the present study, which must be considered when interpreting our findings. First, the camera we used had a field-of-view of 60-degrees, which, whilst comparable to that used in early models [54], does not match current models used in police departments, with the widest recording up to 130-degrees [92]. Whilst a narrower recording range may increase the number of possible 'off-camera' details experienced, we note off-camera details reported were relatively few and believe the design of our study may have resulted in an underestimation of these details. First, the stimulus video was recorded in a single room in which most actors remained. Providing participants rotated fully during the video, their camera would have recorded most of the person and setting details that could be reported. Second, participants were not required to activate their body-worn camera; instead, it recorded the entire duration of the event. Whilst the types of interactions caught on police body-worn cameras may vary widely and inevitably capture interactions where the officer is stationary, we predict that our study provided a narrower scope for events that could be 'missed' by the camera. For example, in research analysing recorded use-of-force incidents, Willits and Makin [93] noted that often the camera was not pointing at the incident or was turned on too late, and examination of multiple videos was often necessary to understand the complete picture. Nonetheless, we still found a small effect of reviewing footage on reporting off-camera details. Future research might investigate late camera activation, shorter or fragmented footage, and increased movement of the individual wearing the camera; all of which we predict would magnify the number of off-camera details reported.

Finally, the current study is limited by the sample. Having a relatively modest sample size limited the power of our analyses, particularly in relation to the critical interaction between the review and statement timing condition. Furthermore, we used a student sample to gather evidence to inform current policing debates. Whilst direct comparisons have not provided consistent evidence for differences in memory between these two groups [94, 95], we acknowledge that police officers' training, reporting style, frequent exposure to crimes, and priorities during an incident (i.e. apprehending suspects or de-escalating threats) may influence what they attend to and recall [96, 97]. Thus, further investigation of the impact of chest-mounted cameras on statement writing and memory within a policing context would inform best practice. It is possible that officers may feel more inclined to produce reports that are identical to their footage due to fears of scrutiny (something we perhaps would expect less of our student participants) [98].

## Conclusion

In conclusion, information in an officer's statement and information shown on their body-worn video are two important components that offer different perspectives on an incident. Contrary to common belief, body-worn video is not always a replica of what an officer experienced. However, an officer's statement may quickly become a replica of what can be seen on their body-worn video due to camera conformity. Our results suggest that in cases in which it is crucial to understand an officer's experience as the action unfolded, an initial statement may be an important first step. Given the undue pressure on officers to give statements that conform to what is on camera, review of footage may otherwise result in the omission of these details.

## Acknowledgments

We are grateful to Natali Dilevski for coding the free recall statements and Elodie Chiarovano for her assistance with virtual reality.

## Author Contributions

**Conceptualization:** Delene Adams, Helen M. Paterson, Hamish G. MacDougall.

**Data curation:** Delene Adams.

**Formal analysis:** Delene Adams.

**Investigation:** Delene Adams, Helen M. Paterson, Hamish G. MacDougall.

**Methodology:** Delene Adams, Helen M. Paterson, Hamish G. MacDougall.

**Project administration:** Helen M. Paterson.

**Resources:** Helen M. Paterson, Hamish G. MacDougall.

**Software:** Hamish G. MacDougall.

**Supervision:** Helen M. Paterson, Hamish G. MacDougall.

**Writing – original draft:** Delene Adams.

**Writing – review & editing:** Helen M. Paterson, Hamish G. MacDougall.

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
