## [Decision Letter · Decision Letter 0]

16 Sep 2020

PONE-D-20-25075

Law and (rec)order: Updating memory for criminal events with body-worn cameras

PLOS ONE

Dear Dr. Paterson,

Thank you for submitting your manuscript to PLOS ONE. After careful consideration, we feel that it has merit but does not fully meet PLOS ONE’s publication criteria as it currently stands. Therefore, we invite you to submit a revised version of the manuscript that addresses the points raised during the review process.

Two reviewers reviewed this manuscript. As you can see from their comments, both have positive things to say about the potential impact of this work. However, both experts, and Reviewer #2 in particular, raise a number of concerns. These comments should be addressed by additional analyses where possible, or should at minimum be taken into account when interpreting and discussing the results. The concern about generalizability of the results to real-life police situations should also receive attention in the manuscript.

We look forward to receiving your revised manuscript.

Kind regards,

Maria Wimber

Academic Editor

PLOS ONE

Journal Requirements:

2. Please provide additional details regarding participant consent. In the Methods section, please ensure that you have specified (1) whether consent was informed and (2) what type you obtained (for instance, written or verbal). If your study included minors, state whether you obtained consent from parents or guardians. If the need for consent was waived by the ethics committee, please include this information.

4. Please upload a copy of Figure 1 and 2, to which you refer in your text on page 11, 18. If the figure is no longer to be included as part of the submission please remove all reference to it within the text.

Reviewers' comments:

Reviewer's Responses to Questions

**Comments to the Author**

1. Is the manuscript technically sound, and do the data support the conclusions?

Reviewer #1: Yes

Reviewer #2: Yes

2. Has the statistical analysis been performed appropriately and rigorously? 

Reviewer #1: Yes

Reviewer #2: Yes

3. Have the authors made all data underlying the findings in their manuscript fully available?

Reviewer #1: Yes

Reviewer #2: Yes

4. Is the manuscript presented in an intelligible fashion and written in standard English?

Reviewer #1: Yes

Reviewer #2: Yes

5. Review Comments to the Author

Reviewer #1: I very much enjoyed reading this manuscript, which describes a single study using VR to explore the completeness and accuracy of witnesses' memory reports of an event, and how both an initial statement and an opportunity to review "Body Cam" footage influences these outcomes.

The study itself is novel and I believe would make a valuable contribution to the literature subject to a few revisions. Whereas the methodology overall is neat, I was especially fascinated by the authors' examination of the discrepancies between the chest-angle and head-angle. I thought this was a really important focus that led to some very interesting findings. The paper overall is well written and the study is positioned very well in the context of existing literature

I have a few suggestions for improvement of the manuscript:

(1) My main concern here is about statistical power, which given the sample size is relatively low for detecting modest effects. I'd suggest that at the least, the authors give this issue some direct treatment in the paper, consider the kinds of effect sizes their study could reliably detect, and moderate some of the claims accordingly.

(2) There are a few places where the wording could be somewhat clearer. In particular (a) the opening few sentences of the introduction; (b) the wording of Hypothesis 3 on p.10, and (c) the paragraph beginning "To assess..." that starts at the bottom of p.17.

(3) The authors might be interested to look at and cite a recent paper by Henkel & Milliken, which seems relevant to their aims and hypotheses: https://doi.org/10.1016/j.jarmac.2020.07.002

(4) I didn't seem to be able to find Figure 1

(5) I was interested in the authors' justification for their control tasks - having people describe the 'mountain' video instead of the crime event, and also having some people review a nature documentary. I wonder whether the inclusion of these control tasks was necessary, and whether they might even have affected the results. In many studies like this, a control task is necessary to avoid the possibility that, for example, the experimental task acts as a distractor, or takes longer. But in this study there seemed no need to control for this possibility. I worry that the control tasks could have artificially weakened performance in the control conditions by preventing rehearsal of the film, for example.

(6) The authors need to include the descriptive data in Tables 2-4 (Table 1 doesn't seem to exist) separated out by initial statement condition (i.e. for each cell in their 2x2 design), not just collapsed across these conditions - regardless of whether the effects were significant.

Reviewer #2: This paper reads rather like a ‘man versus machine’ commentary. As such, the hypotheses are rather obvious, and although I enjoyed reading this work I really struggled with a number of aspects of this project/design, which has left me uncertain as to whether it warrants publication in PlosOne. As such I have recommended a major revision, simply to allow the authors to reply to my concerns. On one hand the research is well conducted, and novel (as far as I am aware) but I am concerned about the applied nature of the literature and tone of the paper: positive but actually rather contentious in terms of not accentuating the importance of confidence and how video footage can undermine one’s confidence, resulting in altering a perfectly good statement of ‘what I remember and what I noticed’ because of what the ‘video says’ and/or ‘doesn’t say’ and the impact of retrieval environment/method.

I liked the use of VR technology. And the paper was well written. A misinformation condition would really improve this work, in my opinion. Readers will find this interesting, and the work is of broad interest.

Concerns

i) The paper is firmly situated I in the context of police and investigation, yet participants are all UG students who clearly differ from police professionals .

ii) The authors have used a written format questionnaire type document to collect the event information, which included some cues, but this deviates from applied practice (in the UK, that is).

iii) I am unsure as to the rationale for having a 1-week delay between encoding and retrieval. I may have missed the rationale for this, but this is at odds with police practice, generally. Please clarify.

iv) The finding ‘analyses revealed a 376significant effect of review condition on the completeness of statements, F(1,93) = 22.72, p377<.001, partial η2= .196, where those who reviewed their body-worn video included more details (both correct and incorrect)’ was in my view to be expected, likewise the accuracy and detail results. The big questions that arise here are, is this improved memory performance, or is this simply a case of altering accounts because video’s don’t lie, so I must be wrong, and/or a loss of confidence in my own memory and not wanting to appear to underperform (which is laid bare given the fact that video footage is available). This research raises this very significant question, but does not in my opinion answer this question.

v) The finding that ‘participants who did not review their body-worn video included a higher proportion of off-camera details in their final statements, M=0.08 (0.10) than participants who reviewed their body-worn video, M= 0.04 (0.06)’ speaks to the above question regarding confidence and altering in line with a ‘video does not lie’ hypothesis. Likewise, ‘a larger proportion of ‘off-camera’ details in their initial statement than in their final statement’. I note that a confidence scale was used, which supports my concerns.

vi) A post event misinformation condition is clearly missing – this would speak to my concerns and make this a more rounded paper.

Additional comments/suggestions

P.4, lines 79-81. Please expand and explain this statement. It is referenced, but I think an additional couple of sentences explaining to readers what the ‘problem’ is would improve this paragraph. In fact, this entire paragraph needs to be unpacked because this paragraph is presenting the rational for this research. The final sentence requires explaining at the very least.

Line 105 - ‘Memories of these details are then subject to further impairment following a period of delay’ the word ‘are’ should be replaced with ‘may’, because this is not always the case.

Line 120 - insert ‘mentally’ return, otherwise this suggests physically return, which is not what is meant here.

Paragraph commencing line 120 needs to be strengthened. The theoretical rational is weak, and should, in my opinion, be better explained and unpacked. At present the theory underpinning the rationale for this research is woolly, and rather tenuous. Suggest looking at the literature on the importance of cues for episodic memory, also. Please add some empirical research citations in this paragraph, rather than relying on books/books chapters, which are quite dated. Spreading activation needs to be explained and fully referenced: the challenges as well as the potential support for your rationale.

The gist versus verbatim literature is also relevant to this paper.

Does body worn camera footage ‘rewrite’ and officer’s memory ? Or does it lead to the officer altering their statement because they begin to doubt their own memory, and loose confidence in their cognition. This is also a question that needs to be considered/highlighted - see lines 177-179.

6. PLOS authors have the option to publish the peer review history of their article (what does this mean?). If published, this will include your full peer review and any attached files.

Reviewer #1: No

Reviewer #2: No

---

## [Author Response · Author response to Decision Letter 0]

29 Oct 2020

Please see attached response letter.

---

## [Decision Letter · Decision Letter 1]

18 Nov 2020

Law and (rec)order: Updating memory for criminal events with body-worn cameras

PONE-D-20-25075R1

Dear Dr. Paterson,

We’re pleased to inform you that your manuscript has been judged scientifically suitable for publication and will be formally accepted for publication once it meets all outstanding technical requirements.

Kind regards,

Maria Wimber

Academic Editor

PLOS ONE

Additional Editor Comments (optional):

As can be seen from the reviewers' comments, both experts feel that their points have been sufficiently addressed in the revision, and recommend the manuscript for publication in its present form.

Reviewers' comments:

Reviewer's Responses to Questions

**Comments to the Author**

1. If the authors have adequately addressed your comments raised in a previous round of review and you feel that this manuscript is now acceptable for publication, you may indicate that here to bypass the “Comments to the Author” section, enter your conflict of interest statement in the “Confidential to Editor” section, and submit your "Accept" recommendation.

Reviewer #1: All comments have been addressed

Reviewer #2: All comments have been addressed

2. Is the manuscript technically sound, and do the data support the conclusions?

Reviewer #1: Yes

Reviewer #2: Yes

3. Has the statistical analysis been performed appropriately and rigorously? 

Reviewer #1: Yes

Reviewer #2: Yes

4. Have the authors made all data underlying the findings in their manuscript fully available?

Reviewer #1: Yes

Reviewer #2: Yes

5. Is the manuscript presented in an intelligible fashion and written in standard English?

Reviewer #1: Yes

Reviewer #2: Yes

6. Review Comments to the Author

Reviewer #1: (No Response)

Reviewer #2: I think this is much improved and very timely paper given recent events in the USA and elsewhere, with a wide appeal. The rational and theory is far tighter, and all my questions regarding the analysis/results have been addressed.

Thank You

7. PLOS authors have the option to publish the peer review history of their article (what does this mean?). If published, this will include your full peer review and any attached files.

Reviewer #1: No

Reviewer #2: No

---

## [Editor Report · Acceptance letter]

24 Nov 2020

PONE-D-20-25075R1 

Law and (rec)order: Updating memory for criminal events with body-worn cameras 

Dear Dr. Paterson:

I'm pleased to inform you that your manuscript has been deemed suitable for publication in PLOS ONE. Congratulations! Your manuscript is now with our production department. 

Kind regards, 

on behalf of

Dr. Maria Wimber 

Academic Editor

PLOS ONE